# Esaxerenone Protects against Diabetic Cardiomyopathy via Inhibition of the Chemokine and PI3K-Akt Signaling Pathway

**DOI:** 10.3390/biomedicines11123319

**Published:** 2023-12-15

**Authors:** Ziyue Li, Huihui Zhang, Weihan Zheng, Zi Yan, Jiaxin Yang, Shiyu Li, Wenhua Huang

**Affiliations:** 1Guangdong Medical Innovation 3D Printing Application Transformation Platform, Third Affiliated Hospital of Southern Medical University, Guangzhou 510630, China; lzyxz112358@163.com (Z.L.); zhengweihan_nick@hotmail.com (W.Z.); jp6303416224@quml.ac.uk (Z.Y.); 2Burns Department, Nanfang Hospital, Southern Medical University, Guangzhou 510515, China; zhh18665052795@163.com; 3Key Laboratory of Medical Biomechanics, Southern Medical University, Guangzhou 510515, China; yjx128397@gmail.com

**Keywords:** diabetic cardiomyopathy, esaxerenone, pharmacological mechanism, therapy, network pharmacology

## Abstract

(1) Background: Diabetic cardiomyopathy (DCM) is a unique form of cardiomyopathy that develops as a consequence of diabetes and significantly contributes to heart failure in patients. Esaxerenone, a selective non-steroidal mineralocorticoid receptor antagonist, has demonstrated potential in reducing the incidence of cardiovascular and renal events in individuals with chronic kidney and diabetes disease. However, the exact protective effects of esaxerenone in the context of DCM are still unclear. (2) Methods: The DCM model was successfully induced in mice by administering streptozotocin (55 mg/kg per day) for five consecutive days. After being fed a normal diet for 16 weeks, echocardiography was performed to confirm the successful establishment of the DCM model. Subsequent sequencing and gene expression analysis revealed significant differences in gene expression in the DCM group. These differentially expressed genes were identified as potential targets for DCM. By utilizing the Swiss Target Prediction platform, we employed predictive analysis to identify the potential targets of esaxerenone. A protein–protein-interaction (PPI) network was constructed using the common targets of esaxerenone and DCM. Enrichment analysis was conducted using Metascape. (3) Results: Compared to the control, the diabetic group exhibited impaired cardiac function and myocardial fibrosis. There was a total of 36 common targets, with 5 key targets. Enrichment analysis revealed that the chemokine and PI3K-Akt signaling pathway was considered a crucial pathway. A target-pathway network was established, from which seven key targets were identified. All key targets exhibited good binding characteristics when interacting with esaxerenone. (4) Conclusion: The findings of this study suggest that esaxerenone exhibits a favorable therapeutic effect on DCM, primarily by modulating the chemokine and PI3K-Akt signaling pathway.

## 1. Introduction

According to statistics, diabetes patients have a significantly elevated risk of developing cardiovascular disease in comparison to the general population. Among diabetes patients, 65% eventually die from cardiovascular disease. Diabetic cardiomyopathy, characterized by myocardial fibrosis, poses a threat to patients with diabetes [1,2]. However, there is currently no effective treatment for diabetic cardiomyopathy (DCM), making it crucial to study its pathogenesis and identify key targets for the prevention and delay of disease progression.

The mineralocorticoid receptor (MR) is a nuclear receptor activated by steroid hormones. A substantial amount of clinical evidence suggests that blocking MRs is an important therapeutic option for heart failure and chronic heart diseases, including complications associated with diabetes [3]. Although there are currently two steroidal MR antagonists, spironolactone and eplerenone, available in clinical practice, their use has not been accepted due to the side effects they can cause. Spironolactone can lead to menstrual irregularities, while eplerenone can cause hyperkalemia. As a result, their clinical use is not yet widely adopted [4].

Esaxerenone, which is a new non-steroidal MR antagonist, improves endothelial dysfunction caused by diabetes by promoting eNOS phosphorylation in the DCM group [5,6]. Several clinical studies have reported that esaxerenone has significant antihypertensive and cardiorenal protective effects compared with spironolactone and eplerenone [7,8]. Pharmacological studies have shown that due to its inverted side-chain structure, esaxerenone has an affinity for MRs that is over 1000 times higher than other NR3C nuclear receptors. Additionally, the binding site of esaxerenone is larger and penetrates deeper into the protein core, making its inhibitory effect more potent and longer-lasting compared with that of spironolactone and eplerenone [9,10]. However, there is still limited research on the impact of esaxerenone on patients with DCM.

In our study, we aimed to establish the DCM model and utilize sequencing and network-pharmacology methods to identify key targets. Additionally, enrichment analysis was conducted to determine the main pathways through which these proteins act in the treatment of DCM with esaxerenone. The aim of this research is to provide treatment evidence for clinical studies.

## 2. Materials and Methods

### 2.1. Experimental Animals

In our study, the mice (male C57BL/6J mice) were housed with a temperature range of 21–23 °C and humidity maintained at 40–60%. The mice were acclimated for 2 weeks before the experiment. They were randomly divided into two groups, with 6 mice in each group. To induce the DCM model, the DCM group mice were treated with streptozotocin (STZ, Sigma-Aldrich, St Louis, MO, USA) at 55 mg/kg per day for 5 days intraperitoneally. STZ was dissolved in citrate buffer, and the control group mice received injections of the vehicle (citrate buffer) for 5 days. One week after the injections, tail vein blood samples were randomly collected from the mice, and their glucose levels were measured from Roche, Indianapolis, IN. Mice glucose levels equal to or higher than 16.7 mmol/L were considered to have successfully developed the diabetic model. Following the establishment of the model, both groups of mice were continuously fed a normal diet for 16 weeks. The mice were anesthetized with 1.5% isoflurane (30 mg/kg) and euthanized. The myocardial tissue was collected for subsequent experiments. Specimens from the middle part of the left ventricle were fixed in 4% paraformaldehyde for histopathological analysis. The remaining left-ventricular tissue was stored at −80 °C, and the cardiac apex tissue was used for RNA sequencing analysis. The protocol was approved by the Experimental Animal Committee of the Third Affiliated Hospital of Southern Medical University, Guangzhou, Guangdong, China. A flow chart of the animal study is shown in Figure 1.

### 2.2. Echocardiographic Assessment

After 16 weeks of establishing the diabetes model, mice were sedated with 1.5% isoflurane and placed in the supine position, and cardiac function was evaluated using the Vevo 2100 system (Fujifilm Visual Sonics, Toronto, ON, Canada). M-mode-and-B-mode echocardiography was performed to measure various parameters related to cardiac function.

### 2.3. Histological Staining

The hearts from two experimental groups, using a 4% paraformaldehyde solution, were embedded in paraffin and sectioned. The sections were then subjected to staining procedures, including Masson’s staining (performed using Sigma-Aldrich, Burlington, MA, USA) and HE staining (performed using Sigma-Aldrich, Burlington, MA, USA). By analyzing the stained sections using ImageJ1.8.0 (Media Cybernetics Inc., Rockville, Maryland, USA), researchers were able to quantify and measure specific features of left-ventricular remodeling, providing insights into the structural changes occurring in the hearts of the experimental group’s RNA sequencing (RNA-seq).

RNA sequencing was performed on two groups of mouse apical cardiac tissues stored at −80 °C. Total RNA was gained using the RNeasy Mini Kit (250) manufactured by Qiagen, Hilden, Germany. Three replicates from both the control and DCM groups underwent quality control and RNA quantification. Strand-specific libraries were prepared and sequenced (Illumina NovaSeq 6000, San Diego, CA, USA). The raw sequencing data underwent initial processing. The sequences were then aligned to reference genes, and differentially expressed genes were identified based on criteria including a *p*-value and a fold change (FC). Differentially expressed genes with FPKM < 1 in each group were excluded.

### 2.4. Data Analysis

The continuous data were reported as mean ± standard error of the mean (SEM). Comparison was performed using a one-way analysis of variance (ANOVA), followed by a post hoc Bonferroni test in case of significant interaction in ANOVA. A *p*-value < 0.05 was considered statistically significant. Statistical analysis was performed using GraphPad Prism 7.0 software (GraphPad Software Inc., San Diego, CA, USA).

### 2.5. Network Pharmacology

#### 2.5.1. Potential Targets of Esaxerenone

The 2D structure of esaxerenone was obtained from PubChem and gained the targets of esaxerenone from the Swiss Target Prediction platform (http://www.swisstargetprediction.ch/, accessed on 21 August 2023) [11]. Targets with a probability greater than 0.1 were filtered out to refine the prediction results. All target names were standardized according to Uniprot (http://www.uniprot.org/, accessed on 21 August 2023) [12].

#### 2.5.2. Construction of PPI Network

We utilized an online tool called Interactive Venn to generate a Venn diagram, where the overlapping regions represent the typical targets of esaxerenone and DCM [13]. The identified targets were then uploaded to the STRING 11.0 platform (http://string-db.org/, accessed on 25 August 2023), and a protein–protein-interaction (PPI) network was generated. This PPI network provided a comprehensive view of the potential protein interactions and functional associations among the identified targets [14].

#### 2.5.3. Enrichment Analysis

The Metascape platform was used to perform enrichment analysis (https://metascape.org/, accessed on 15 September 2023) [15]. The top-20 Kyoto Encyclopedia of Genes and Genomes (KEGG) pathways (*p* < 0.05) and top-10 Gene Ontology (GO) items were selected for analysis. The results were further processed using EHBIO for visualization.

#### 2.5.4. Construction of Target-Pathway Network

We uploaded the results of enrichment analysis to Cystoscopes 3.7.2 [16] to identify the core targets for intervention in DCM. Cystoscopes 3.7.2 revealed the interactions between components, targets, and pathways, providing insights into potential therapeutic targets.

#### 2.5.5. Molecular Docking Verification

The 2D structure information of esaxerenone was gained from the PubChem database [17]. The candidate targets were found on the PDB database to download their 3D structures [18]. Autodock Vina was employed for saving pdbqt-format files, which are compatible with molecular docking software [19]. The molecular docking verification was visualized using PyMOL 2.4.025 and Discovery Studio 2019 [20]. A flow chart of the network-pharmacology study is shown in Figure 1.

## 3. Results

### 3.1. Myocardial Injury in DCM Mice

Normal and diabetic-cardiomyopathy mice were used to evaluate cardiac function (Figure 2A). Through echocardiography (Figure 2B), the contraction and relaxation functions of the mouse heart were evaluated. The DCM group showed notable impairment in ventricular contraction function compared to the control group, manifested by a decrease in both fractional shortening (FS%) and ejection fraction (EF%). Additionally, the DCM group showed poorer stroke volumes (SV), end-systolic left-ventricular internal diameter (LVESD), and end-systolic left-ventricular volume (LVESV), indicating the successful establishment of the DCM mouse model (Figure 2C).

Tissue samples were collected from the mice and subjected to HE staining and Masson’s staining. Figure 2D demonstrates that the cardiomyocytes in the DCM group exhibited significant sparsity and hypertrophy, while the myocardial fibers appeared fragmented and disorganized. Additionally, Figure 2E,F reveal that collagen fiber deposition was enhanced in the DCM group (*p* < 0.0001) compared to the control group, as observed through Masson’s trichrome staining.

### 3.2. Candidate Targets of Esaxerenone and DCM

In the RNA sequencing analysis of root tip tissue in the DCM group, a total of 24,197 genes were detected, among which 375 genes showed differential expression, with 74 upregulated genes and 301 downregulated genes (Figure 3A). Through screening of literature and online databases, a total of 104 candidate genes for esaxerenone were identified. A total of 36 common genes were identified between esaxerenone and DCM (Figure 3B). Then, a heatmap was generated using these common genes (Figure 3C).

### 3.3. PPI Network

In order to explore the target genes’ functions and identify connections in complex diseases, we submitted the overlapping targets to the STRING website to construct a PPI. After removing disconnected nodes, the resulting network consisted of 32 nodes and 106 edges. The topological parameters were analyzed, and the 32 targets were ranked based on their degrees. They were then arranged into concentric circles, as shown in Figure 3D. To gain further insights into the mechanism of esaxerenone in treating DCM, MCODE was used to perform cluster analysis. This analysis identified a potential protein functional module with the highest score, as depicted in Figure 3E. Proteins within such a module are believed to have closer relationships and may interact with each other to perform specific biological functions. Therefore, these targets were predicted to be important, including PIK3CA, PIK3CB, CDK4, JAK3, CCR2, CCR3, and CXCR3. These targets have the potential to play crucial roles in the mechanism of action of esaxerenone in the treatment of DCM.

### 3.4. Enrichment Analysis

To gain insights into the biological effects of esaxerenone related to the treatment of DCM, we conducted GO functional analysis and KEGG-pathway enrichment analysis. Figure 4 displays the top-10 GO terms and top-20 KEGG pathways, which were selected based on their *p*-values.

For biological processes, the targets were mainly enriched in the chemokine-mediated signaling pathway, phosphatidylinositol-3-phosphate biosynthetic process, peptidyl-threonine phosphorylation, and amyloid-beta formation. For cellular components, the enrichment analysis revealed that the targets were predominantly associated with protein kinase activity, kinase activity, and protein tyrosine kinase activity. In terms of molecular functions, the targets were primarily enriched in receptor complexes, phosphatidylinositol 3-kinase complexes, extrinsic components of the membrane, and glutamatergic synapses. The KEGG-pathway analysis indicated that the majority of the pathways involved were associated with the chemokine signaling pathway, PI3K-Akt signaling pathway, neurotrophin signaling pathway, and ErbB signaling pathway. Notably, the chemokine signaling pathway exhibited a considerable concentration of targets, emphasizing the significance of the chemokine-mediated signaling pathway in the context. Additionally, biological processes involved the chemokine-mediated signaling pathway and phosphatidylinositol-3-phosphate biosynthetic process. The cellular component involved phosphatidylinositol kinase activity, and molecular functions involved the phosphatidylinositol 3-kinase complex. Therefore, we thought that the chemokine signaling pathway and PI3K-Akt signaling pathway should be important.

### 3.5. Compound-Pathway Network

To establish a target-pathway network, we connected potential pathways and hub genes based on the results of the KEGG analysis. This network provides a visual representation of the relationships between the enriched pathways and the genes involved (Figure 5). Among the important targets identified through the PPI network, the average degree of connectivity for the included targets was 7.36. PIK3CA (Degree = 20), PIK3CB (Degree = 20), CDK4 (Degree = 7), and JAK3 (Degree = 7) were enriched in the PI3K-Akt signaling pathway, and CXCR3 (Degree = 7), CCR2 (Degree = 7), and CCR3 (Degree = 7) were enhanced in the chemokine signaling pathway.

### 3.6. Molecular Docking Result Analysis

In this study, we performed molecular docking to validate the interaction between the core genes and fibrosis. It is commonly accepted that a lower energy value indicates a more stable conformation of the ligand–receptor binding and a higher likelihood of interaction. We conducted docking simulations, and the majority of the binding complexes showed strong binding affinities, with an average energy of −7.14 kcal/mol. The modes of eight binding complexes were displayed in Figure 6, including esaxerenone-PI3KCA(7bi4) docking (−8.0 kcal/mol), esaxerenone-CXCR3(6wzl) docking (−5.9 kcal/mol), esaxerenone-PI3KCB(4v0i) docking (−6.7 kcal/mol), esaxerenone-CDK4(3g33) docking (−7.5 kcal/mol), esaxerenone-JAK3(6dap) docking (−6.3 kcal/mol), esaxerenone-CCR2(5t1a) docking (−6.6 kcal/mol), and esaxerenone-CCR3(7x9y) docking (−7.7 kcal/mol). Using the esaxerenone-PI3KCA docking as an example, it is observed that the small-molecule ligand esaxerenone potentially binds to the interface pocket created by the interaction of protein amino acid residues (Figure 6A(a)). As shown in Figure 6A(b), a hydrogen bond was formed between esaxerenone and THRB:471, near the active site of PI3KCA. The other essential residues (TYRB:470, GLNB:475, GLNB:475, SERB:474, ASNA:467, LYSA:678, HOHA:1296, ASNA:465, HISA:450, TYRB:467, PROA:449, PROA:447, TRPA:424, VALA:448, TBPA:446, THRA:678, GLYA:1009, ASNA:677, and HISA:676) interacted with esaxerenone through van der Waals forces, pi-alkyl interaction, alkyl, carbon–hydrogen bond, and water hydrogen bond. For esaxerenone-CXCR3, the essential residues (HOHD:349, ARGD142, GLUD:143, HOHD:337, PROD:141, HOHD:303, GLND:199, TYRD140, GLUD:105, LYSD:107, SERD:12, THRD:10, PROD:8, and LEUD:11) interacted with esaxerenone through van der Waals, water hydrogen bond, conventional hydrogen bond, carbon–hydrogen bond, unfavorable donor–donor, amide-pi stacked, alkyl, pi-alkyl, and halogen (fluorine). For esaxerenone-PI3KCB, the essential residues (LYSN:598, PHEB:595, ARGB:601, SERB:594, GLUB:628, CYSB:590, METB:387, CYSB:569, LEUB:488, TYRB:591, ASPB:626, and CYSB:627) interacted with esaxerenone through van der Waals, conventional hydrogen bond, carbon–hydrogen bond, halogen (fluorine), unfavorable donor–donor, pi-sulfur, alkyl, and pi-alkyl. For esaxerenone-CDK4, the essential residues (TYRA:170, ALAA:21, TYRA:22, ARGA:186, ASPA:145, VALA:190, ARGA:144, LEUA:183, GLUA:224, VALA:181, and PROA:178) interacted with esaxerenone through van der Waals, conventional hydrogen bond, halogen (fluorine), Pi-Anoon alkyl, and pi-alkyl. For esaxerenone-JAK3, the essential residues (PROA:896, SERA:860, GLNA:896, LEUA:898, PHEA:868, LEUA:857, GLNA:856, ARGA:899, GLUA:819, and GLNA:858 interacted with esaxerenone through van der Waals, conventional hydrogen bond, carbon–hydrogen bond, halogen (fluorine), unfavorable donor–donor, pi-sulfur, alkyl, and pi-alkyl. For esaxerenone-CCR2, the essential residues (PROA:896, LEUA:133, ILEA:217, LEUA:213, ILEA:132, PHEA:129, VALA:167, LEUA:209, ILEA:163, THRA:164, THRA:160, VALA:159, and PHEA:156 interacted with esaxerenone through van der Waals, halogen (fluorine), pi-sigma, alkyl, and pi-alkyl. For esaxerenone-CCR3, the essential residues (PROA:896, LEUA:133, ILEA:217, LEUA:213, ILEA:132, PHEA:129, VALA:167, LEUA:209, ILEA:163, THRA:164, THRA:160, VALA:159, and PHEA:156 interacted with esaxerenone through van der Waals, halogen (fluorine), pi-sigma, alkyl, and pi-alkyl. Hydrogen bonds and other types of interactions played a vital role in facilitating the stable binding of small molecules to the active sites of their target proteins. These interactions were crucial for maintaining the structural integrity and functional specificity of the binding complexes.

## 4. Discussion

In individuals with diabetes, DCM stands as a prominent contributor to both heart failure and mortality, and currently, there is no recognized treatment for it. The development of DCM is influenced by factors such as increased secretion in mineralocorticoid hormones due to high blood glucose and insulin resistance. The current treatment methods include ACE inhibitors/ARBs [21], guanosine monophosphate cyclohydrolase stimulators, sodium-glucose co-transporter 2 inhibitors [22], MR antagonists, or modulation of T-cells [23,24,25,26]. These treatments can reduce the incidence and mortality rate. Currently, the use of non-steroidal MR antagonists is being evaluated for the treatment of heart failure, both as standalone therapy and in combination with sodium-glucose co-transporter 2 inhibitors. These groundbreaking drugs have the potential to become important therapies for various cardiac and renal diseases. Esaxerenone has shown promising cardiovascular effects in patients with diabetes. However, additional investigation is needed to pinpoint the precise treatment targets and mechanisms associated with DCM. In our study, we utilized the network-pharmacology method to predict and elucidate the potential molecular mechanisms of action of esaxerenone in DCM.

To delve into the core constituents and action mechanism of esaxerenone in DCM treatment, a PPI network was established for 36 therapeutic targets. Utilizing topology analysis on the PPI network, a total of seven key pathogenic genes were identified, namely PIK3CA, PIK3CB, CDK4, JAK3, CCR2, CCR3, and CXCR3. Among them, PIK3CA and PIK3CB correspond to the genes encoding the alpha subunit of phosphatidylinositol 3-kinase. Frequent mutations in PIK3CA and PIK3CB have been identified in patients with DCM. These mutations have been found to be associated with myocardial fibrosis and cardiac dysfunction [27]. Additionally, studies have shown that excessive activation of PIK3CA and PIK3CB may lead to increased myocardial cell proliferation and inflammation, exacerbating the pathological process of DCM [28]. Playing a pivotal role in governing the progression of the cell cycle, particularly in cellular proliferation and division, CDK4 emerges as a vital protein [29]. Excessive activation of CDK4 has been implicated in the development of myocardial fibrosis and cardiac dysfunction. Research has demonstrated that inhibiting CDK4 can attenuate the extent of myocardial fibrosis and improve cardiac function in animal models of DCM [30]. JAK3 (Janus kinase 3) is a member of the Janus kinase family of proteins that play a crucial role in signal transduction pathways involved in inflammation and immune responses [31]. Activation of the JAK/STAT pathway has been observed in various cardiovascular diseases, including DCM, and is associated with inflammation, oxidative stress, and fibrosis, which are key pathological processes in DCM [32]. Inhibition of JAK/STAT signaling has shown promising results in attenuating cardiac dysfunction and fibrosis in animal models of DCM. Although the exact role of JAK3 in DCM is not fully elucidated, targeting the JAK/STAT pathway, including JAK3, has emerged as a potential therapeutic strategy [33]. Both CCR2 and CCR3 are chemokine receptors that play a role in the recruitment and activation of immune cells in inflammatory conditions [34]. This upregulation is associated with increased infiltration in monocytes, macrophages, and eosinophils into the cardiac tissue, leading to chronic inflammation. CCR2-and-CCR3-mediated inflammatory responses actively contributes to the generation of pro-inflammatory cytokines, oxidative stress, and fibrosis, all of which represent significant pathological characteristics observed in DCM [35]. CXCR3 is another chemokine receptor that plays a role in immune cell recruitment and activation. Our studies have shown that CXCR3 is upregulated in the hearts of diabetic animals and patients with DCM. This upregulation is associated with increased infiltration in immune cells, such as T-cells and macrophages, into the cardiac tissue [36]. The activation of CXCR3 leads to the release of pro-inflammatory chemokines, such as CXCL9 and CXCL10, which further promote the recruitment of immune cells and contribute to the chronic inflammation observed in DCM [37,38].

Among the various pathways analyzed using KEGG-pathway enrichment analysis, the chemokine and the PI3K-Akt signaling pathway emerged as the most prominent, displaying significant roles in relation to the study of interest (Figure 7). Furthermore, by constructing a compound-pathway network, it was observed that all seven targets were significantly enriched in both the chemokine and the PI3K-Akt signaling pathway. This reaffirms the potential efficacy of esaxerenone treatment in DCM by highlighting the favorable impact of the chemokine signaling pathway and the PI3K-Akt signaling pathway. So far, there is considerable clinical evidence to suggest that esaxerenone, as an MR inhibitor, can significantly improve the prognosis of patients with diabetes [39,40,41]. The chemokine signaling pathway is significant in the pathogenesis of DCM. Chemokines are small signaling proteins that regulate the migration and activation of immune cells [42,43]. In DCM, chronic inflammation and immune cell infiltration contribute to the development and progression of cardiac dysfunction and fibrosis. The involvement of the PI3K signaling pathway in cardiac pathology is well established, and changes in the expression and activity of PI3K have a significant influence on the progression of diabetic cardiomyopathy [44,45]. Research findings have demonstrated that esaxerenone exhibits the potential to decrease the occurrence and fatality rate of diabetic nephropathy and cardiovascular events, enhance myocardial hypertrophy and interstitial fibrosis, and slow down the pathological advancement of left-ventricular systolic dysfunction [39,46]. Similarly, in patients with diabetic cardiomyopathy, MR inhibitors not only prevent left-ventricular remodeling and expansion of fibrotic areas but also protect against myocardial injury [46].

### Limitations of the Study

This study solely relied on network pharmacology to predict the molecular mechanisms of MR antagonist treatment for DCM, specifically involving chemokines and the PI3K-Akt signaling pathway. The main limitation of this study is the lack of experimental validation to verify the accuracy and reliability of these predictions. While previous research has established a clear association between the pathogenesis of heart failure and the overexpression of MR and aquaporin-1, supported by relevant clinical studies, the mechanism through which the benefit of inhibitors of MR is still unclear for DCM [47,48]. Additionally, we are aware that the guanosine monophosphate (GMP) cyclohydrolase pathway plays a role, but the pathway we predicted did not enrich the GMP pathway.

However, from the perspective of practicing physicians, these research findings can still provide some guidance for clinical practice. Firstly, esaxerenone can be considered one of the drug choices for treating diabetic cardiomyopathy. Clinicians can incorporate esaxerenone into treatment regimens and personalize the treatment based on individual patient characteristics. Secondly, chemokines and the PI3K-Akt signaling pathway may be the target of esaxerenone’s effects. Therefore, clinicians can focus on the activity of these signaling pathways and related biomarkers when evaluating patients with diabetic cardiomyopathy. This can help better understand the patient’s condition and provide more accurate evidence for treatment plan formulation. Finally, we believe that it is necessary to experimentally determine the pathway through which this drug treats diabetic cardiomyopathy, and this will be the focus of our next study.

## 5. Conclusions

In our study, we utilized a DCM model and employed RNA sequencing and network-pharmacology approaches to investigate the molecular mechanisms underlying the therapeutic effects of esaxerenone in DCM treatment. By analyzing a complex network, we identified key targets and highlighted the involvement of the chemokine and PI3K-Akt signaling pathway in mediating the therapeutic effects of esaxerenone. Our initial predictions also indicated promising interactions between esaxerenone and targets within these pathways. These preliminary findings provide scientific evidence supporting the potential clinical application of esaxerenone in DCM treatment. Further research is warranted to explore these findings in more detail.

## Figures and Tables

**Figure 1 biomedicines-11-03319-f001:**
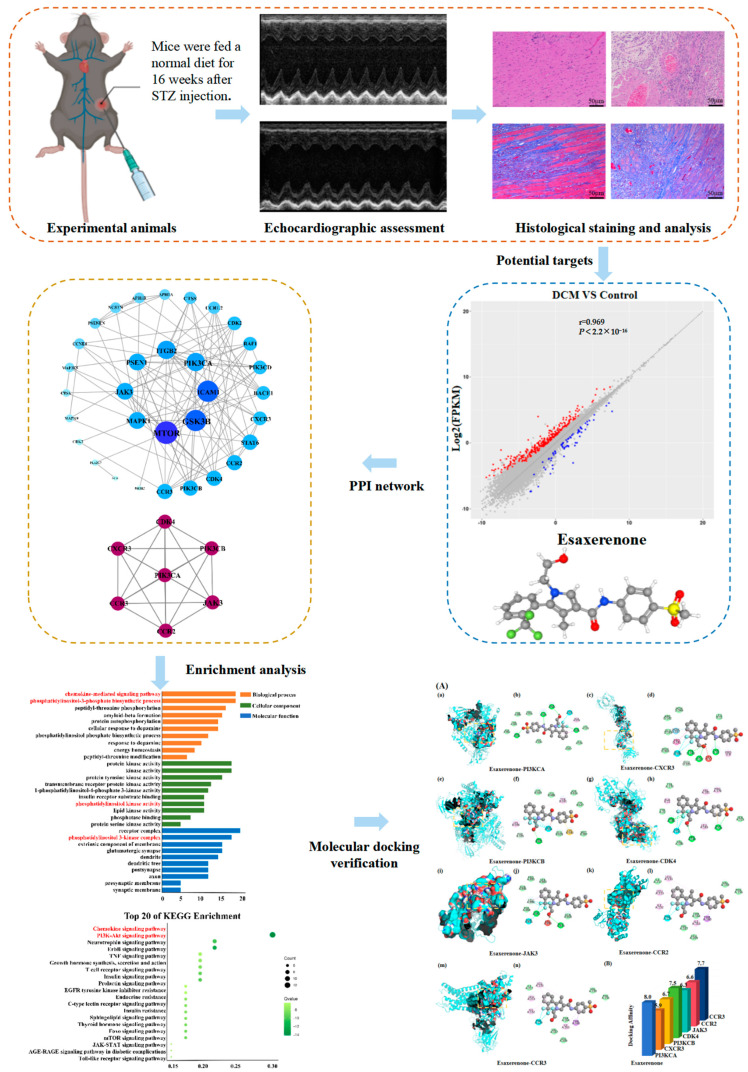
Scheme of investigating esaxerenone in treatment of diabetic cardiomyopathy.

**Figure 2 biomedicines-11-03319-f002:**
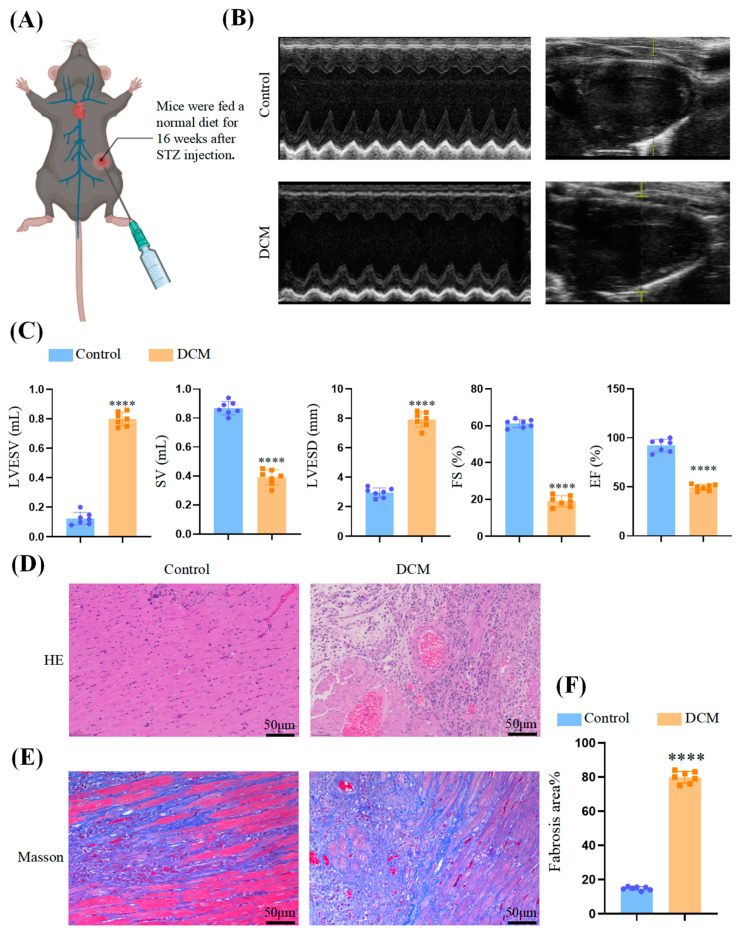
Echocardiographic variables and cardiac fibrosis in mice. (**A**) Experimental animals. (**B**) The quantitative results of echocardiography. (**C**) Results are shown as the mean ± SEM (n = 6). Significance: **** *p* < 0.0001. (**D**) HE staining in 2 groups. (**E**) Masson’s trichrome staining in 2 groups. (**F**) Measurements of fibrotic area in cardiac cross sections in 2 groups of mice.

**Figure 3 biomedicines-11-03319-f003:**
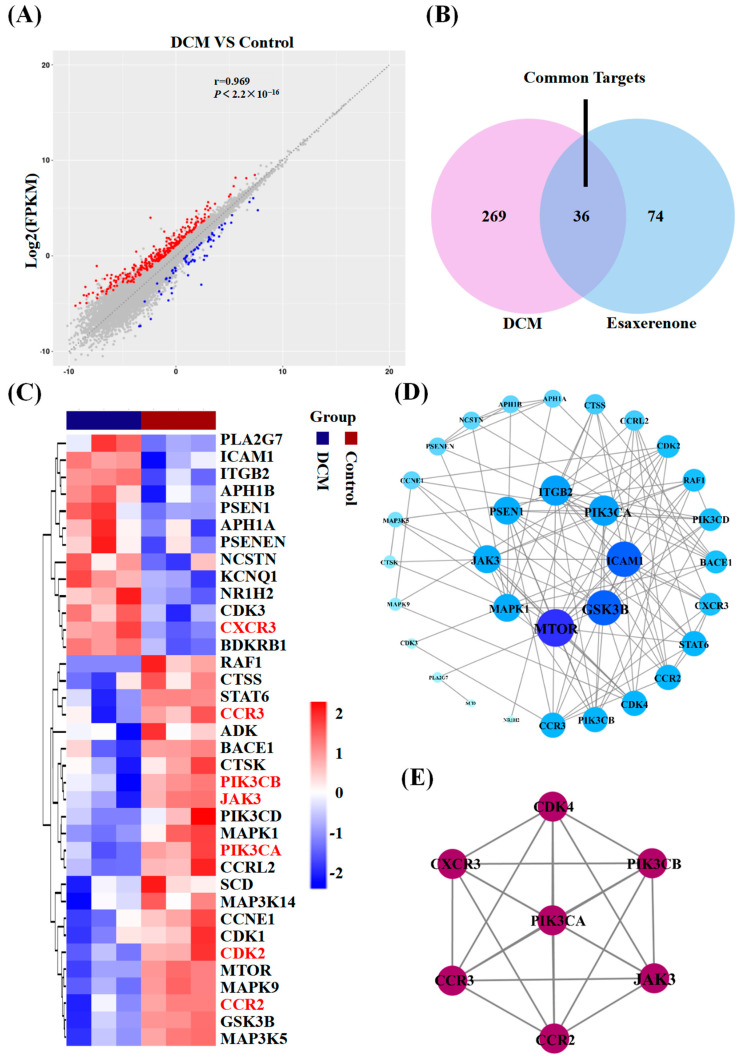
Common targets of esaxerenone in the treatment of DCM. (**A**) The scatter plot displays the gene expression differences in 2 groups. The horizontal and vertical axes represent the samples from the control and DCM groups, respectively. Upregulated genes in the DCM group are shown in red, downregulated genes in blue, and insignificant genes in gray. (**B**) The Venn diagram illustrates the overlap between DCM-related targets and esaxerenone-related targets. The pink section represents DCM-related targets, while the blue section represents esaxerenone-related targets. There are 36 common targets in the overlapping section. (**C**) The heat map visualizes the differential-gene-expression patterns. Each row represents a differential gene, each column represents a mouse sample, and each group consists of 3 replicates. (**D**) The PPI network consists of 32 target proteins and 106 interacting edges. The node sizes reflect the degree values, with larger nodes indicating higher degrees of connectivity. (**E**) The PPI network analysis identified a potential protein functional module with the highest score.

**Figure 4 biomedicines-11-03319-f004:**
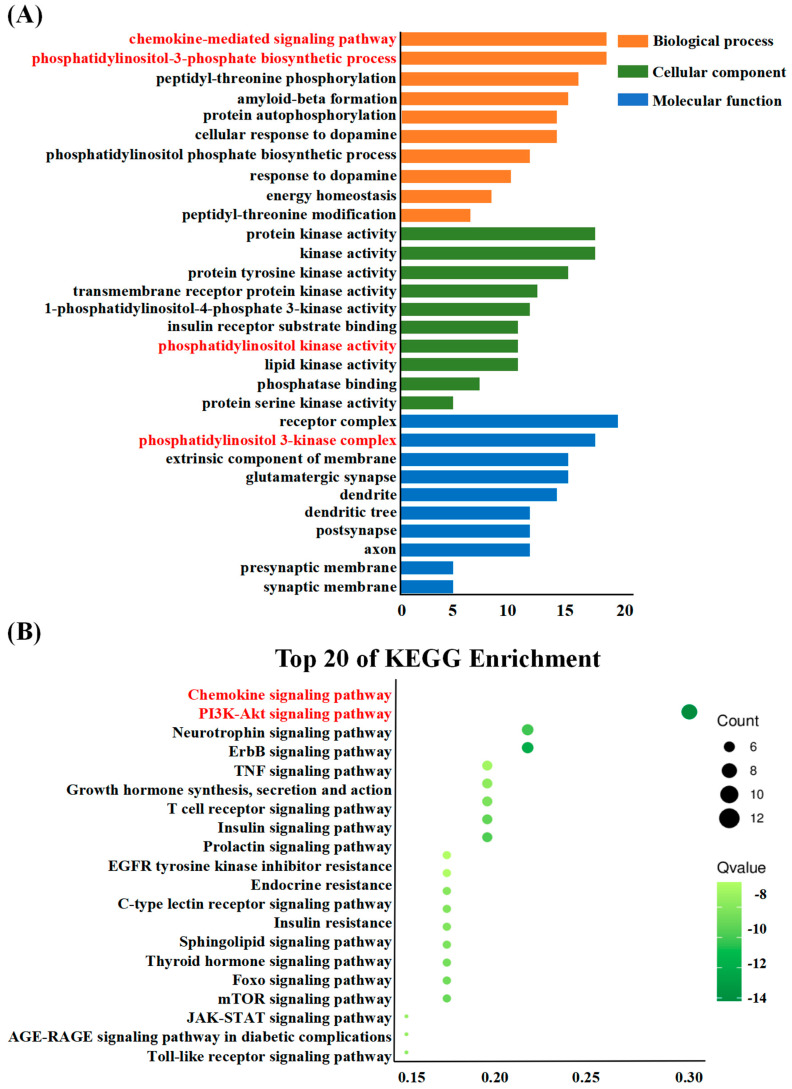
Enrichment analysis. (**A**) GO functional analysis. Biological process items in GO analysis enriched for up- and downregulated genes. (**B**) The KEGG-pathway enrichment analysis represents the sizes and color of the bubbles, indicating the potential targets associated with each pathway.

**Figure 5 biomedicines-11-03319-f005:**
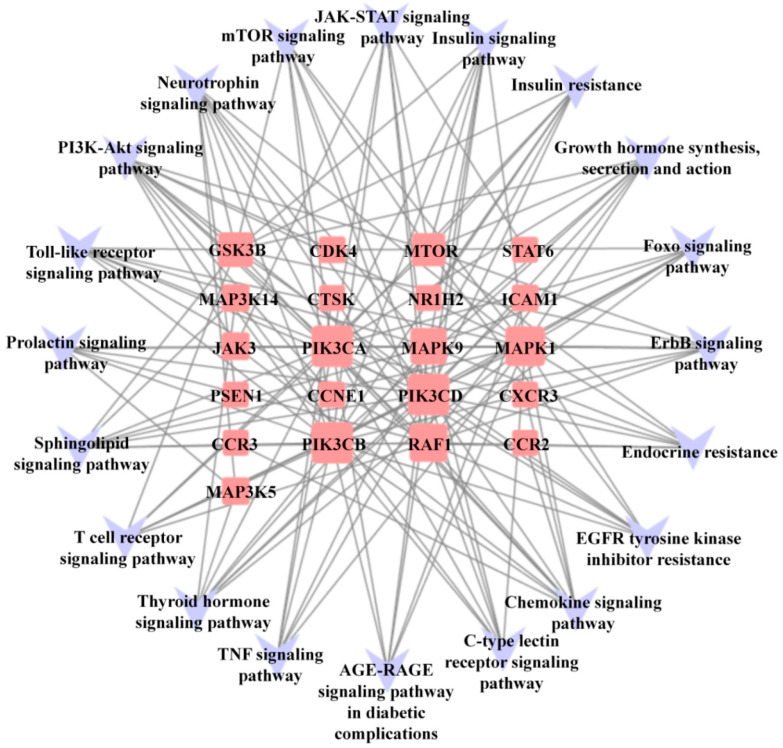
The compound-pathway network has 41 points and 151 lines. The targets are represented by pink squares, while the pathways are represented by 20 purple V-shapes. The sizes of the square nodes are arranged in degree values, from large to small.

**Figure 6 biomedicines-11-03319-f006:**
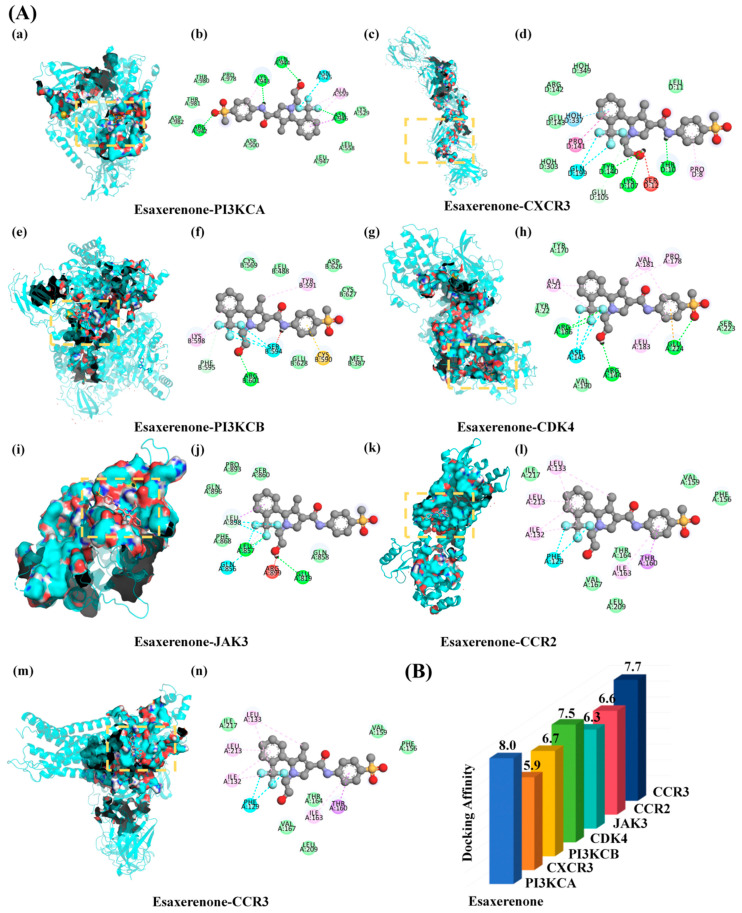
Molecular docking. (**A**) The diagram displays conformations obtained from the molecular docking simulation. The 3D models illustrate esaxerenone’s molecular structure within the protein’s binding pocket, represented as orange sticks. The surrounding amino acid residues are depicted in a surface style. The interactions between the esaxerenone and the neighboring residues are visually depicted in the 2D diagrams. (**B**) The docking affinity of the column diagram.

**Figure 7 biomedicines-11-03319-f007:**
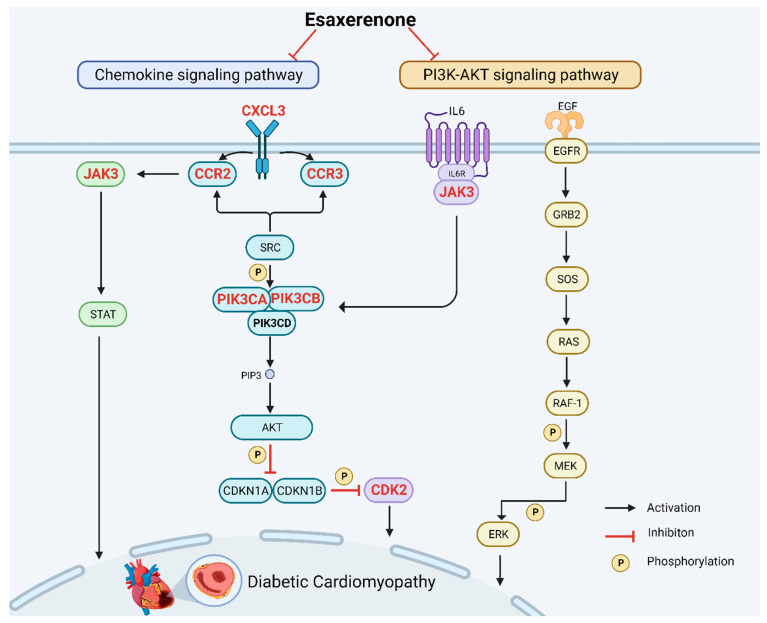
Chemokine signaling pathway and PI3K-Akt signaling pathway influenced by esaxerenone. The red nodes represent the hub genes; the other nodes are the genes of pathway.

## Data Availability

The data presented in this study are available on request from the corresponding author.

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
