# Peer review of "Esaxerenone Protects against Diabetic Cardiomyopathy via Inhibition of the Chemokine and PI3K-Akt Signaling Pathway"

_biomedicines, 2023, doi:10.3390/biomedicines11123319_

Round 1

Reviewer 1 Report

Comments and Suggestions for Authors

The authors analyzed the mechanisms of action of esoxarenon in an animal model of dilated cardiomyopathy (DCM). A very important topic from a practical point of view. Understanding the mechanisms will be an element that will improve the effectiveness and safety of therapy from a clinical point of view. The study was designed and conducted correctly. Results clearly presented, well discussed.
In order to increase the quality of work, I propose to introduce several extensions.
- the most important therapeutic pathway in DCM patients is the guanylate cyclase-cGMP pathway. Most drugs currently used in DCM act directly or indirectly on this pathway. Please add the relationship between this pathway and esoxarenone and take into account the role of the youngest class of nitric oxide and carbon monoxide mediators, as well as peptide mediators in these pathways.
- In the authors' opinion, is such an effect a class effect (I mean non-steroidal mineralocorticoid receptor antagonists) or drug-specific? If specific, what distinguishes this molecule? Can the results justify differences in registration indications (finerenone - prevention of progression of chronic diabetic kidney disease (and small effect on blood pressure) and esoxarenon - arterial hypertension?
- the conclusion should contain instructions not only summarizing the study but also emphasizing possible directions of therapeutic action - from the point of view of a practicing physician. I would like to ask you to expand the presented conclusions.

Author Response

Q1:In order to increase the quality of work, I propose to introduce several extensions.

- the most important therapeutic pathway in DCM patients is the guanylate cyclase-cGMP pathway. Most drugs currently used in DCM act directly or indirectly on this pathway. Please add the relationship between this pathway and esoxarenone and take into account the role of the youngest class of nitric oxide and carbon monoxide mediators, as well as peptide mediators in these pathways.

A1:Thanks to your comments, which we added on Page 15, line 388-399

Q2:In the authors' opinion, is such an effect a class effect (I mean non-steroidal mineralocorticoid receptor antagonists) or drug-specific? If specific, what distinguishes this molecule? Can the results justify differences in registration indications (finerenone - prevention of progression of chronic diabetic kidney disease (and small effect on blood pressure) and esoxarenon - arterial hypertension?

A2:These two drugs are similar in that they both act through the salicorticoid receptor, and I think that the research done now can only guide the clinical use of esoxarenon, and that the research on finerenone and esoxarenon is something that needs to be the focus of our next research on this topic.

Q3: the conclusion should contain instructions not only summarizing the study but also emphasizing possible directions of therapeutic action from the point of view of a practicing physician. I would like to ask you to expand the presented conclusions.

A3:Thanks to your comments, which we added on Page 16, lines 399-410

Reviewer 2 Report

Comments and Suggestions for Authors

The article entitled, “Esaxerenone Protects Against Diabetic Cardiomyopathy via Inhibition of the Chemokine and PI3K-Akt Signaling Pathway." is a good read. This study aims to investigate the protective effects of Esaxerenone, a selective nonsteroidal mineralocorticoid receptor antagonist, during diabetic cardiomyopathy. 

In this study DCM model was induced in mice by administering streptozotocin (55 mg/kg per day) for five consecutive days with 16 weeks on a normal diet, echocardiography confirmed the successful establishment of the DCM model. Subsequent gene expression analysis identified significant differences in gene expression in the DCM group, pinpointing potential targets for DCM. Utilizing the Swiss Target Prediction platform, they performed predictive analysis to identify potential targets of esaxerenone. Also, a protein-protein interaction (PPI) network, formed by common targets of esaxerenone and DCM, was constructed. The authors discovered in comparison to the control group, the group of individuals with diabetes demonstrated compromised heart function and the presence of myocardial fibrosis. The results of the enrichment analysis indicated that both the chemokine signaling pathway and the PI3K-Akt signaling pathway were identified as significant pathways. This study found that esaxerenone improves DCM by altering the chemokine and PI3K-Akt signaling pathways.

Altogether this is an important and timely article, this reviewer has certain suggestions that would help produce a more comprehensive overview of the topic: 

Comments:

1, Replace the word “Introduction” form abstract (line 19) to “Background”.

2, The English of manuscript can be polished (minor) and there are few typological errors. Do rectify them.

3, Authors can add one paragraph for abbreviations.

4, Authors should add a paragraph to discuss limitations of this study.

5, Role of other immune cells are also very important factor in heart diseases, therefore I would suggest adding few citations to put comprehensive view of this topic (PMID: 35730443; PMID: 36465455; PMID: 27494688 PMID: 34119620 etc.)  

6, Add Statistical analysis in materials and methods section.  

7, Fig. 1 quality may be improved (high resolution).

Comments on the Quality of English Language

Minor editing of English language required

Author Response

Q1, Replace the word Introduction form abstract (line 19) to Background.

A1:Thanks to your comments, we have changed it to Background ".

Q2: The English of manuscript can be polished (minor) and there are few typological errors. Do rectify them.

A2:Thank you for your suggestion, we have made detailed changes

Q3: Authors can add one paragraph for abbreviations.geo dcm stz

A3:Thanks to your comments, which we added on Page 15, lines 426-435

Q4, Authors should add a paragraph to discuss limitations of this study.

A4:Thanks to your comments, which we added on Page 15, line 387-398

Q5: Role of other immune cells are also very important factor in heart diseases, therefore I would suggest adding few citations to put comprehensive view of this topic (PMID: 35730443; PMID: 36465455; PMID: 27494688 PMID: 34119620 etc.)  

A5:Thanks to your comments, we have cited the literature at Page 14, line 320

Q6: Add Statistical analysis in materials and methods section.  

A6:Thanks for your comment, we added in Page 3, line128-123

Q7: Fig. 1 quality may be improved (high resolution).

A7:Thanks for your comment,Fig. 1 quality has been improved

Q8:Comments on the Quality of English Language Minor editing of English language required

A8:Thank you for your comments, we have made changes

Reviewer 3 Report

Comments and Suggestions for Authors

In this paper the authors addressed the topic of diabetic cardiomyopathy (DCM) going deeply in the use of a mineralocorticoid receptor antagonist, esaxerenone, for the pathology treatment. In particular, using a preclinical mouse model of DCM, they utilize sequencing and network pharmacology approaches to better elucidate the mechanism of action of esaxerenone.

Authors provided at first, results of the mouse model characterization and then the network analysis.

One observation is that methods related to the animal model and its characterization are poorly described and they need to be implemented for a better comprehension. Moreover, many mistakes along the text and in figures and their legends are present, leading to the conclusion of a very superficial review of the text before submission.

Major and minor points are reported in the list here below.

Major points:

1-    Methods related to animal model preparation and characterization are poorly described. As examples, the methods related to animal sacrifice and heart explantation are completely missing and it is reported that RNA is gained, but no explanation about the method is provided. A revision and implementation is needed.

2-    A lot of inaccuracies are present along all the text, in figures and relative legends. A complete revision is needed.

Minor points:

1-    Page 2, line 59: the word “it” is not necessary in the sentence.

2-    Page 2, line 81: which is the route of streptozotocin injection? Intravenous, intraperitoneally,…? Please specify in the text.

3-    Page 2, line 86: as reported in the abstract, please specify that mice were fed with normal diet.

4-    Page 2, line 91: the trademark of Vevo should be at apex.

5-    Page 2, line 92: were should be corrected in was.

6-    Page 3, line 94: the histology section does not report the protocol for animal sacrifice and heart explantation. Please provide this information before the description of the histological procedure.

7-    Page 3, line 95-96: the sentence lack the governing verb. A “were” before embedded is needed.

8-    Page 3, line 102: the method to obtain the RNA is missing. From which samples, with which process?

9-    Page 3, line 125: the sentence lack the governing verb.

10- Page 4, Figure 1: the reported figure is not cited and commented in the text. Moreover the article “the” should be removed from the legend. To better report the experimental workflow on the mice, a further image between streptozotocin treatment and echocardiographic assessment should be added with the indication of the 16 days normal diet period.

11- Page 5, Figure 2: same comment of Figure 1 regarding the introduction of a further image explaining the complete experimental protocol on mice. Moreover, the legend should be revised. The general title is about echocardiographic variables, but the figure report also on histological analysys. The D panel is only related to HE staining. The E panel refers to Masson and consequently an F panel reports measurement of fibrotic area. Please correct the legend accordingly.

12- Page 5, Figure 3: the description of panel B in the legend reports finerenone-related targets. Please correct in esaxerenone.

13- Page 7, line 165: the word function should be corrected in functions.

14- Page 7, line 169: authors report “poorer SV, LVID and ESCV. Looking at graphs in Figure 2, this is true for SV, but LVID and ESCV are higher in DCM group than in the control one.

15- Page 7, line 172: tissue samples were collected from mice and not from rats. Please correct.

16- Page 7, line 175: as reported for Figure 2 legend, panels should be revised. Panel E refers to Masson images and an updated panel F refers to measurement of fibrotic area.

17- Page 7, line 201: dapagliflozin is cited in the treatment of DCM. This should be corrected in esaxenerone.

18- Page 7, line 210: please substitute capital letter in The with lowercase.

19- Page 8, line 248: please substitute capital letter in Esaxenerone with lowercase.

20- Page 9, Figure 4: if the sense of targets written in red is to highlight the main ones as described in the text, not all are reported in this way. Same consideration for the top 20 of KEGG enrichment. Moreover the description of panel A in the legend is not exhaustive, because not only biological processes are reported.

21- Page 13, line 322: “significant significance” sounds not so good. Please modify the sentence.

Comments on the Quality of English Language

Minor revisions are needed.

Author Response

Q1: Page 2, line 59: the word it is not necessary in the sentence.

A1:Thank you for your feedback. Page 2, line 58: We have removed 'it'."

Q2: Page 2, line 81: which is the route of streptozotocin injection? Intravenous, intraperitoneally,? Please specify in the text.

A2: Your suggestion is very accurate. Page 2, line 81: It is stated that it was administered via intraperitoneal injection

A3: Page 2, line 86: as reported in the abstract, please specify that mice were fed with normal diet.

A3: Your suggestion is very true , Page 2, line 86 shows that it is a normal diet.

Q4: Page 2, line 91: the trademark of Vevo should be at apex.

A4: Thank you for your comments, Page 3, line 97, we have made changes to the logo.

Q5:  Page 2, line 92: were should be corrected in was.

A5: Thank you for your comment, Page 3, line 98, we have changed were to was

Q6:  Page 3, line 94: the histology section does not report the protocol for animal sacrifice and heart explantation. Please provide this information before the description of the histological procedure.

A6: Thank you for your comment, Page 2, line 86, we have made the change.

Q7:  Page 3, line 95-96: the sentence lack the governing verb. A were before embedded is needed.

A7:Thank you for your comment, Page 2, line 102, we have added were" before embedded.

Q8: Page 3, line 102: the method to obtain the RNA is missing. From which samples, with which process?

A8:Thank you for your comment, Page 3, line 108, we have made the change.

Q9: Page 3, line 125: the sentence lack the governing verb.

A9: I'm sorry, we didn't find that management verb missing, if you can, could you please hint at it?

Q10: Page 4, Figure 1: the reported figure is not cited and commented in the text. Moreover the article the should be removed from the legend. To better report the experimental workflow on the mice, a further image between streptozotocin treatment and echocardiographic assessment should be added with the indication of the 16 days normal diet period.

A10:Thank you for your comments, we have made the changes Page 5, Figure 1.

Q11:Page 5, Figure 2: same comment of Figure 1 regarding the introduction of a further image explaining the complete experimental protocol on mice. Moreover, the legend should be revised. The general title is about echocardiographic variables, but the figure report also on histological analysys. The D panel is only related to HE staining. The E panel refers to Masson and consequently an F panel reports measurement of fibrotic area. Please correct the legend accordingly.

A11:Page 6, Figure 2, we have made changes based on your suggestions.

Q12:Page 5, Figure 3: the description of panel B in the legend reports finerenone-related targets. Please correct in esaxerenone.

A12:Page 5, Figure 3,we have made changes based on your suggestions.

Q 13:Page 7, line 165: the word function should be corrected in functions.

A13:Thank you for your comments, we have made the changes.

Q14: Page 7, line 169: authors report poorer SV, LVID and ESCV. Looking at graphs in Figure 2, this is true for SV, but LVID and ESCV are higher in DCM group than in the control one.

A14:Thank you for your comments, we believe that LVESV, LVESD in heart disease early myocardial decompensation myocardium can not be fully contracted, so in the early stage is going up, and in the late stage is going down.

Q15: Page 7, line 172: tissue samples were collected from mice and not from rats. Please correct.

A15:Thank you for your comments, we have made changes Page 7, line 188.

Q16: Page 7, line 175: as reported for Figure 2 legend, panels should be revised. Panel E refers to Masson images and an updated panel F refers to measurement of fibrotic area.

A16:Thank you for your comments, we have made changes Page 8, line 191.

Q17: Page 7, line 201: dapagliflozin is cited in the treatment of DCM. This should be corrected in esaxenerone.

A17:Thank you for your comments, we have made changes Page 8, line 214

Q18: Page 7, line 210: please substitute capital letter in The with lowercase.

A18:Thank you for your comments, we have made changes Page 8, line 222.

Q19: Page 8, line 248: please substitute capital letter in Esaxenerone with lowercase.

A19:Thank you for your comments, we have made changes Page 9, line 263.

Q20: Page 9, Figure 4: if the sense of targets written in red is to highlight the main ones as described in the text, not all are reported in this way. Same consideration for the top 20 of KEGG enrichment. Moreover the description of panel A in the legend is not exhaustive, because not only biological processes are reported.

A20:Thank you for your comments, which we have explained in detail Page 9, line 232-236.

Q21:Page 13, line 322: significant significance sounds not so good. Please modify the sentence.

A21:hank you for your comments, we have made changesPage 15, line 368.

Reviewer 4 Report

Comments and Suggestions for Authors

The manuscript by Li and co-authors describes the investigation of esaxerenone influence on diabetic cardiomyopathy. The authors performed a systematic study involving the experimental and modelling methodologies. It was discovered that esaxerenone causes a favorable therapeutic effect on diabetic cardiomyopathy, presumably via the chemokine and PI3K-Akt signaling pathway. The results are important and deserve publication.

Specific comments:

Page 3, Section "Molecular docking verification":  The authors indicated that the ligand and protein were saved in PDBQT format using AutoDock Tools. However, it is not clear, which protonation scheme was applied. Also, the program used for the docking calculations should be mentioned (AutoDock, AutoDock Vina, or other program?) along with the program options. 

Figure 1:  Some symbols in the figure are too small for the chosen resolution.

Page 8, Lines 232-243:  What was a source of the protein structures for docking? If Protein Data Bank was searched for the structures, then the PDB codes should be indicated. Also, it is not clear, which molecule was docked (esaxerenone?). In Figure 6 (Panel A) structures of different ligands are shown. For example, the ligands under (b), (f), and (j) are different. Please mention them in the text of the manuscript and in caption of Figure 6.

Lines 244-252:  The binding mode of esaxerenone with just one biotarget (CCR3) was described. Why the description of binding modes with other biotargets was omitted here?

Figure 6:  The labels of amino acid residues are hardly visible.

Summarizing, I recommend major revision of the manuscript before publication.

Author Response

Q1: Page 3, Section "Molecular docking verification":  The authors indicated that the ligand and protein were saved in PDBQT format using AutoDock Tools. However, it is not clear, which protonation scheme was applied. Also, the program used for the docking calculations should be mentioned (AutoDock, AutoDock Vina, or other program?) along with the program options. 

A1: Under normal physiological conditions PH = 7.4. we used AutoDock Vina for molecular docking that has been modified in page3.

Q2: Figure1:  Some symbols in the figure are too small for the chosen resolution.

A1:We've made adjustments to figure1.

Q3:Page 8, Lines 232-243:  What was a source of the protein structures for docking? If Protein Data Bank was searched for the structures, then the PDB codes should be indicated. Also, it is not clear, which molecule was docked (esaxerenone?). In Figure 6 (Panel A) structures of different ligands are shown. For example, the ligands under (b), (f), and (j) are different. Please mention them in the text of the manuscript and in caption of Figure 6.

A3:The protein structure is derived from the PDB database, which we have supplemented with PDB codes, we are using esaxerenone for docking, and due to our mistake of not unifying the esaxerenone structure, changes have been made to the figure6 ligands.

Q4:Lines 244-252:  The binding mode of esaxerenone with just one biotarget (CCR3) was described. Why the description of binding modes with other biotargets was omitted here?

A4:We have added a description of all biotargets with esaxerenone.

Q5:Figure 6:  The labels of amino acid residues are hardly visible.

A5:We have made changes to figure6.

Round 2

Reviewer 1 Report

Comments and Suggestions for Authors

the authors have made corrections, I believe that the work may be considered for publication

Author Response

Thank you very much for your recognition. I still made some modifications to my article, please review

Reviewer 3 Report

Comments and Suggestions for Authors

Authors have answered to questions in a proper way. Just few clarifications here below, one of them requested by the authors:

1- Page 4 line 136: the sentence without a governing verb is: ""Enrichment analysis using the Metascape platform". This is a telegram , not a sentence.

2- Authors have performed implementation of Figures 1 and 2 adding the information of "injection of STZ after 16 weeks of normal diet". This is something different from the text were it is reported that STZ was injected, then 1 week later there was blood sampling and then mice experienced 16 weeks of normal diet before sacrifice. Please clarify and eventually modify figures or text accordingly to the right protocol.

3- Figure 2:  authors have changed the legend, but there is still a mistake. Panel D refers to H&E, Panel E refers to Masson's trichrome and panel F to measurement of fibrotic area. Please correct accordingly.

Author Response

Q1:Page 4 line 136: the sentence without a governing verb is: "Enrichment analysis using the Metascape platform". This is a telegram , not a sentence.

A1:Your suggestion is very accurate. Page 4, line 136: “Enrichment analysis using the Metascape platform” the sentence has been modified as "Metascape platform was used to perform enrichment analysis".Page 4 line 136

Q2:Authors have performed implementation of Figures 1 and 2 adding the information of "injection of STZ after 16 weeks of normal diet". This is something different from the text were it is reported that STZ was injected, then 1 week later there was blood sampling and then mice experienced 16 weeks of normal diet before sacrifice. Please clarify and eventually modify figures or text accordingly to the right protocol.

A2:Thank you for your review. We have corrected this error, and we have modified the modeling method in Figure 1 and Figure 2 to "Mice were fed a normal diet for 16 weeks after STZ injection."

Q3:Figure 2: authors have changed the legend, but there is still a mistake. Panel D refers to H&E, Panel E refers to Masson's trichrome and panel F to measurement of fibrotic area. Please correct accordingly.

A3:Thank you for pointing out our mistake. I have corrected the captions, and the revised captions are as follows:(D) HE staining in 2 groups. (E) Masson's trichrome staining in 2 groups. (F) Measurements of fibrotic area in cardiac cross sections in 2 groups of mice.

Reviewer 4 Report

Comments and Suggestions for Authors

The revised version of the manuscript was significantly improved by the authors. I recommend acceptance of the manuscript for publication in the revised form.

Author Response

(The authors gave the same response as above.)
